# The Target of Rapamycin Signalling Pathway in Ageing and Lifespan Regulation

**DOI:** 10.3390/genes11091043

**Published:** 2020-09-03

**Authors:** Ivana Bjedov, Charalampos Rallis

**Affiliations:** 1UCL Cancer Institute, Paul O’Gorman Building, 72 Huntley Street, London WC1E 6DD, UK; 2School of Life Sciences, University of Essex, Wivenhoe Park, Colchester CO4 3SQ, UK

**Keywords:** ageing, *Drosophila*, yeast, nutrient-response, autophagy, translation, transcription, metabolism, TORC1, TORC2

## Abstract

Ageing is a complex trait controlled by genes and the environment. The highly conserved mechanistic target of rapamycin signalling pathway (mTOR) is a major regulator of lifespan in all eukaryotes and is thought to be mediating some of the effects of dietary restriction. mTOR is a rheostat of energy sensing diverse inputs such as amino acids, oxygen, hormones, and stress and regulates lifespan by tuning cellular functions such as gene expression, ribosome biogenesis, proteostasis, and mitochondrial metabolism. Deregulation of the mTOR signalling pathway is implicated in multiple age-related diseases such as cancer, neurodegeneration, and auto-immunity. In this review, we briefly summarise some of the workings of mTOR in lifespan and ageing through the processes of transcription, translation, autophagy, and metabolism. A good understanding of the pathway’s outputs and connectivity is paramount towards our ability for genetic and pharmacological interventions for healthy ageing and amelioration of age-related disease.

## 1. Introduction

The nutrient-sensing and evolutionarily conserved mechanistic target of rapamycin (mTOR) signalling pathway is a pro-ageing signalling hub regulating stress, growth and metabolism from yeast to human [1,2,3]. The centrepiece of the pathway is the phosphatidylinositol 3-kinase-related TOR serine/threonine kinase, firstly discovered in budding yeast, that associate with two (in most eukaryotes), structurally and functionally distinct, multiprotein complexes termed TORC1 and TORC2 [1,2,3,4]. TORC complexes coordinate a plethora of basic cellular organisation and metabolism processes (Figure 1A) such as transcription, translation, autophagy, metabolism, as well as cytoarchitecture and have multiple interactions with other pathways, for instance the Insulin Growth Factor (IGF) and AMP-activated protein kinase (AMPK) signalling pathways [3,5,6].

TORC1 is shown to control global translation through two distinct mechanisms. Firstly, via direct phosphorylation of the translation regulators eukaryotic initiation factor 4E-binding protein 1 (4E-BP1) [7,8] and ribosomal protein S6 kinase 1 (S6K1) [9,10,11,12,13]. Secondly, through regulation of transcription (dependent on all three RNA polymerases) of genes coding for ribosomal proteins [1,9,14], ribosome biogenesis factors [15] and tRNAs [16,17,18,19,20]. Apart from translation-related genes TORC1 has major positive impact on gene expression related to growth and metabolism [2,18] and a negative action on genes coding for proteins related to cellular stress [21,22,23]. Genetic or pharmacological inhibition of mTOR activity stimulates autophagy, a conserved lysosomal catabolic pathway controlling the degradation and turnover of macromolecules and organelles and maintaining metabolic homeostasis at both the cellular and whole-organism level [24,25]. On the other hand, TORC2 has reverse but coordinated roles to TORC1 [4,26] and is implicated, among others, in transcription, cell survival, DNA damage, telomere length, gene silencing, stress response, cytoskeletal organisation lipogenesis and glucose transport [6,27,28,29,30,31]. TORC2 is activated in both a growth factor-dependent and -independent fashion [32]. These effects are mediated through phosphorylation of AGC kinase family members such as AKT, SGK1 and PKC [32]. Traditionally, TORC1 has been associated with temporal while TORC2 with spatial aspects of cellular growth. Nevertheless, recent evidence shows that such distinctions have become somewhat obscure [3]. Both TORC1 and TORC2 complexes and their upstream regulators such as the TSC1–TSC2 (hamartin and tuberin, respectively) protein complex that represses TORC1 by affecting Rheb, a G-protein that acts as positive regulator of this complex [33], show great conservation within eukaryotes (Figure 1B).

In most organisms, mTOR signalling can be inhibited by rapamycin, a macrolide from *Streptomyces hygroscopicus* bacteria living within the soil on the Rapa Nui or Easter Island [34,35]. Rapamycin exhibits broad anti-proliferative properties and is a potent anti-tumour and immunosuppressant drug [36]. Rapamycin directly binds FKBP12 and the complex then binds and inhibits the TOR kinase [37] with beneficial effects on both health span and lifespan of all cellular and organismal systems tested [5]. Intense research, so far, has shown that genetic or pharmacological interventions that lower mTOR activity result in upregulation of stress responses, alteration of mitochondrial respiration and metabolism, slower ageing rates and prolongment of lifespan [5,18,22,25,38,39]. In this review, we briefly present a snapshot of roles of the mTOR pathway in ageing and lifespan regulation, focusing on four basic cellular processes: gene transcription, protein translation, autophagy and metabolism presenting specific examples. Accumulating evidence and increasing understanding of the mechanisms of mTOR-related processes in ageing will help towards the design of interventions that will increase health span and ameliorate mTOR-dependent age-related conditions.

## 2. Lifespan Effects of mTOR through Regulation of Gene Transcription and Metabolism

mTOR exerts many of its effects through the regulation Pol I, Pol II and Pol III transcription realms [5,40,41]. For example, TORC1 regulates autophagy partly through phosphorylation and inactivation of the nuclear translocation of the transcription factor EB (TFEB) which controls genes for lysosomal biogenesis and the autophagy machinery [42,43,44]. Recent work in *Drosophila* has identified REPTOR and REPTOR-Binding Protein (REPTOR-BP) as transcription factors downstream of TORC1 required for ∼90% of the transcriptional induction that occurs upon TORC1 inhibition [45]. However, a REPTOR homologue or functionally related transcription factor in mammals and yeast is yet to be identified. The localisation of mTOR kinases and complexes is the focus of a recent review [46]. Here, we concentrate on examples of mTOR-dependent ageing-related transcriptional effects in diverse model systems from yeast to human cells (Figure 2).

### 2.1. GATA Factors, Maf1 and tRNA Gene Regulation Downstream of mTOR

TORC1 is thought to mediate part of the life-extending effects of dietary restriction [47]. Systematical analysis of transcription via RNA-seq across organs of *Drosophila*, subjected to dietary restriction or lowered mTOR and subsequent prediction of transcription regulators, identified the evolutionarily conserved GATA factors [48]. Lifespan and fitness analyses showed that knockdown of the GATA transcription factor *srp* in specific fly tissues recapitulated the benefits but not the costs of dietary restriction [48]. These results indicated that GATA factors mediate effects of dietary amino acids on lifespan and placing them as target for longevity interventions [49]. GATA factors have been linked to ageing downstream of mTOR in *Caenorhabditis elegans.* ChIP-seq data screening towards identification of transcription factors that regulate to age-related genes identified *elt-2*, a gene responsible for inducing expression of intestinal genes during embryogenesis. Interestingly expression of ELT-2 protein decreases during ageing, persistent ELT-2 expression is a common feature of long-lived animals [50] and is found to be required for TORC1-dependent lifespan extension in hypoxic conditions [51].

Nitrogen is a limiting element required for fuelling cellular growth and anabolic processes. Nitrogen utilization is employed by the alleviation of Nitrogen Catabolite Repression (NCR) which ensures the use of non-preferential or alternative nitrogen sources when preferential sources are not available. In yeasts, TORC1 has been linked with nitrogen catabolite repression and GATA-dependent transcription. When cells are cultured under nitrogen rich conditions TORC1 activation results in Gln3 and Gat1 cytoplasmic sequestration and transcription of genes required to scavenge poor nitrogen sources are highly repressed. Pharmacological or nutritional repression of TORC1 leads to PP2A-dependent translocation of Gln3 and Gat1 to the nucleus [52]. Fission yeast Gat1 orthologue, Gaf1, has similar roles. TORC1 phosphorylates and inhibits Gaf1 and upon TORC1 down-regulation Gaf1 is dephosphorylated by the PP2A-like phosphatase Ppe1 [53]. This allows it to enter the nucleus where it upregulates amino acid permeases including *isp7* (coding for an oxygenase controlling amino acid permeases) and negatively regulates *ste11* (coding for a transcription factor implicated in mating and sporulation) [54,55]. Upregulated genes in *gaf1* mutant cells grown in nitrogen-rich medium are enriched in nitrogen starvation-responsive genes Ste11 targets [54]. While budding yeast GATA transcription factors are not directly implicated in lifespan regulation, a recent study [18] indicated that, similarly to the *C. elegans elt-2*, the fission yeast gaf1 is an anti-ageing gene. Cells lacking *gaf1* are short-lived and Gaf1 is partially required for Torin1-induced lifespan extension [18]. Gene expression, DNA binding and downstream analyses showed that Gaf1 defines a transcriptional control of protein translation mechanism, tuning the production of components required for protein translation according to mTORC1 activity. Importantly, the study showed that Gaf1 binds and directly suppresses all tRNA genes [18] and mediates both Pol II- and Pol III-dependent transcription downstream of mTORC1.

Inhibition of Pol III, which transcribes the tRNAs, prolongs lifespan in yeast, worms, and flies, and is required for the lifespan extension mediated by TORC1 inhibition [40]. Pol III transcription is regulated by a plethora of basic transcription factors (such as TFIIIB, TFIIIC, or TBP) and several other factors without direct DNA binding [56], including the coactivator PNRC as well as MYC which interacts with the Pol III basal apparatus [57,58,59] and the conserved Pol III inhibitor Maf1. The latter is negatively regulated via phosphorylation by the mTOR pathway and tailors 5S rRNA and tRNA production in response to various environmental cues [60]. Recent data have shown that the Maf1 binding site on Pol III overlaps with that of TFIIIB in the preinitiation complex [61]. Deletion of *maf1* in the budding yeast *Saccharomyces cerevisiae* but not its homologue *mafr-1* in *C. elegans* prevents dietary restriction or caloric restriction to extend lifespan. Interestingly, *mafr-1* deletion increases stress tolerance and extends lifespan due to an enhancement of stress response, including oxidative stress response, mitochondrial unfolded protein response (UPRmt) and autophagy [62]. Recent data in fission yeast suggest that Maf1-dependent inhibition of tRNA synthesis controls lifespan by preventing genomic instability that arises at tRNA genes [63]. Overall, numerous studies provide evidence that GATA factors regulate a multitude of metabolic processes downstream of mTOR through the action of all three polymerases.

### 2.2. Metabolic Homeostasis and Lipid Synthesis through ATF4 and SREBPs

The general amino acid signalling pathway harmonises availability of amino acids with protein translation (discussed below). Amino acid starvation leads to TORC1 activity repression and lifespan extension. In the transcriptional level, following TORC1 inactivation, the expression of the leucine zipper transcription factor Gcn4 (in yeast) [64] or ATF4 (activating transcription factor 4) in mammalian cells [65,66] is induced. ATF4/Gcn4 induces amino acid transporters and metabolic enzymes [67,68,69,70,71,72], as well as autophagy factors [73]. Interestingly TORC1 can be inhibited through ATF4-dependent mechanisms upon leucine, arginine, or glutamine deprivation with the mechanism in long-term deprivation including SESTRIN2 [74,75]. On the other hand, TORC1 activation leads to *de novo* purine synthesis through the ATF4-dependent expression of MTHFD2, a bifunctional enzyme and key component of the mitochondrial tetrahydrofolate cycle that provides one carbon units for purine synthesis [76]. Purine metabolism has recently been highlighted as a longevity target with dietary sugar intake strongly predicting circulating purine levels in humans. In addition, high-sugar diets promote accumulation of uric acid, an end-product of purine catabolism [77] and regulating uric acid production impacts on lifespan in a water-dependent manner [77]. Synthesis of pyrimidines is also promoted by TORC1, via its down-stream effector S6K, which phosphorylates and activates CAD (carbamoyl-phosphate synthetase 2, aspartate transcarbamylase, and dihydroorotase), a rate limiting enzymes in the first steps of pyrimidine *de novo* synthesis [78,79].

Sterol regulatory element-binding proteins (SREBPs) are key basic helix–loop–helix leucine zipper (bHLH-Zip) transcription factors that govern fat metabolism [80]. They are synthesized as inactive precursors bound to the endoplasmic reticulum (ER) [81]. Mammalian genomes have two SREBP genes, SREBP-1, and SREBP-2. SREBP-1 has two distinct isoforms, SREBP-1a and -1c, differing in their first exons. SREBP-1a is a potent activator of all SREBP-responsive genes in proliferating cells, including those that mediate the synthesis of cholesterol, fatty acids, and triglycerides [82]. SREBP-1c regulates *de novo* lipogenesis and plays a major role in the nutritional regulation of fatty acid and triglyceride (TG) synthesis in lipogenic organs, such as the liver, while SREBP-2 ubiquitously regulates sterol synthesis in tissues [83]. Upon its activation, mTORC1 promotes lipid synthesis through SREBPs contributing to cellular and organ growth, although the connections of the TORC1 and TORC2 complexes with the activation of SREBPs might be more complicated as crosstalk with other pathways that regulate mTOR, such as the AMP-activated protein kinase (AMPK), are involved [84]. The roles of SREBPs in ageing are not clear yet: results using tissue culture models suggest that enhanced lipogenesis via SREBP-1 induction is involved in senescence [85]. Blocking lipogenesis with fatty acid synthase (FAS) inhibitors and via siRNA-mediated silencing of SREBP-1 and ATP citrate lyase attenuated H_2_O_2_-induced senescence. In addition, aged human fibroblasts were effectively reversed to young-like cells by challenging with FAS inhibitors [85]. Calorie restriction in rats enhances fatty acid (FA) biosynthesis, and SREBP-1c, a master regulator of FA synthesis, has been identified as a mediator of some of the beneficial effects of calorie restriction [86]. *SREBP-1c* knockout mice do not exhibit extended lifespan following calorie restriction and fail to upregulate factors involved in FA biosynthesis. SREBP-1c is shown to be implicated in calorie restriction-associated mitochondrial activation through the upregulation of peroxisome proliferator-activated receptor γ coactivator-1α (PGC-1α), a master regulator of mitochondrial biogenesis [86]. In *C. elegans*, the homolog of mammalian SREBP-1 (sbp-1) in complex with MDT-15 (mediator-15) prevents life-shortening effects of a glucose-rich diet. The upregulation of the SREBP-1/MDT-15 transcription factor complex was necessary and sufficient to mitigate the detrimental effects of a glucose-rich diet to lifespan. SREBP-1/MDT-15 were able to reduce the levels of saturated fatty acids and moderated glucose toxicity on lifespan [87]. Finally, triple-drug combinations in nematode (that included mTOR inhibition) and effectively prolonged lifespan and improved health span, resulted in increased levels of monounsaturated fatty acids (MUFAs) in phosphatidylethanolamines (PEs) and phosphatidylcholines (PCs) membrane lipids [88]. The increase in MUFAs, as well as the synergistic lifespan benefits, required sbp-1 (SREBP-1) [88]. Overall, the data so far suggest interesting connections of SREBP transcription factors with lifespan and dietary restriction downstream of mTOR. Further studies might highlight SREBP-related interventions in age-related metabolic syndromes.

### 2.3. Hypoxia and HIF-1 in Lifespan Regulation Downstream of mTOR

Studies in diverse systems have shown that HIF1, a heterodimeric transcriptional complex containing HIF-1α and HIF-1β, is a target of the TOR pathway [89,90,91,92]. Increased mTOR and S6K activities enhance HIF-1 levels under both normoxic and hypoxic conditions [89,91,93,94]. Under hypoxia, HIF-1α is stabilized promoting cells adaptation to low-oxygen stress [95]. HIF-1 overexpression is expressed in solid tumours and inhibition of HIF-1 can prevent tumour growth [96,97]. However, TORC1 is reported to control growth by shifting cells from oxidative phosphorylation to glycolysis, a phenomenon occurring in cancer cells, through increase of HIF1α translation which drives the expression of several glycolytic enzymes [98].

In *C. elegans*, studies have shown that HIF-1 can modulate lifespan both positively and negatively. For instance, stabilization of HIF-1 increases lifespan and health span through a mechanism distinct from the insulin-like signalling pathway and from dietary restriction [99,100]. HIF-1 was found to act in parallel to the SKN-1/NRF and DAF-16/FOXO transcription factors to promote longevity [99]. On the other hand, HIF-1 is reported to modulate longevity in a nutrient-dependent manner and the *hif-1* loss-of-function mutant extends lifespan under rich nutrient conditions but fails to show lifespan extension under dietary restriction [101]. In addition, inactivation of the *maa-1* (membrane-associated acyl-CoA binding protein 1), which is a *C. elegans* paralogue of ACBP (Acyl-CoA-binding protein), promotes lifespan extension and resistance to different types of stress. HIF-1 is required for the anti-ageing response induced by MAA-1 deficiency. This response relies on the activation of molecular chaperones known to contribute to maintenance of the proteome [102]. The discrepancy between studies could be explained through temperature. Indeed, stabilization of HIF-1 increased life span under all conditions but deletion of *hif-1* increased lifespan at 25 °C but not at 15 °C [103]. At these low temperatures, RNAi knockdown of *hif-1* impaired health span due to age-dependent loss of vulval integrity. Interestingly, *hif-1* knockdown resulted in nuclear localisation of the DAF-16 FOXO transcription factor that was found necessary for the *hif-1*-dependent life span extension at all temperatures [103].

In *Drosophila*, HIF-1α expression is increased in diapause-destined pupal brains and mitochondrial biogenesis is negatively regulated through HIF-1α. This effect is mediated via c-Myc activity inhibition through proteasome-dependent degradation of c-Myc. The mitochondrial transcription factor A (TFAM), encoding a key mitochondrial factor for transcription and DNA replication, is activated through direct binding of c-Myc on TFAM promoter. The HIF-1α-c-Myc-TFAM signalling pathway is, therefore, shown to participate in the regulation of mitochondrial activity for insect diapause or lifespan extension [104]. Changes in HIF-1α with age have been reported in rats and could be related to age-related pathologies [105], such as neurodegeneration [106].

HIF-1α regulates mitochondrial biogenesis, and modulation of the nuclear-mitochondrial communication during aging depends on PGC-1α [107]. Notably, TORC1 balances energy metabolism through transcriptional control of mitochondrial oxidative function via the YY1/PGC-1α transcriptional complex required for rapamycin-dependent repression of respiratory genes [108]. HIF-1α is found to accumulate in the nucleus of SIRT1-silenced primary myoblasts and adult SIRT1 knockout mice [109]. The increased HIF-1α levels observed during ageing or in response to SIRT1 knockout activated *Mxi1*, encoding a c-Myc transcriptional repressor. Mxi1 restricts the interaction between c-Myc and mitochondrial transcription factor A (TFAM) that is critical for replication, transcription, and maintenance of mitochondrial biogenesis [110]. *SIRT1* transcription is also coupled with hypoxia during ageing and during acute hypoxia, HIF-1α and HIF-2α interact with Hsp90, and directly bind to hypoxia response elements in the SIRT1 promoter [111]. Overall, numerous studies support the implication of HIF factors in lifespan and their connections with mitochondrial biogenesis and stress response.

## 3. Lifespan Regulation through mTOR Effects on Protein Translation

One of the principal effects of the mTOR pathway is adjusting protein synthesis and proteome content, according to available nutrients and growth factors. mTOR exerts one of its most prominent, mechanistic links to ageing through regulation of translation (Figure 3), which is described below.

During protein synthesis, mRNAs are scanned for start codon in an interaction with different complexes. The 43S preinitiation complex (PIC) is formed from 40S ribosomal subunit, initiation factors eIF3, eIF1, and eIF5, while a ternary complex consists of methionyl-initiator tRNA (Met-tRNAi) and GTP-bound eIF2. The m^7^G cap structure, situated at the 5′ end of mRNAs, is recognised by eIF4F cap-binding complex. An important feature of eIF4F complex, formed of eIF4E (cap binding), eIF4A (RNA helicase) and eIF4G (scaffolding protein), is its very low cellular concentration, which places it in a critical position to regulate translation [112]. The cap structure of mRNA is also a binding site for different RNA-binding proteins, which can shield them from translation [113]. Some mRNAs, typically from viruses and likely circular mRNAs as well, bypass mRNA scanning by using highly structured internal ribosomal entry site (IRES) elements to recruit PIC to 5′ UTR [114,115,116,117,118]. Once the AUG start codon is encountered in the 5′ UTR within a favourable sequence context, GTP is hydrolysed and release of eIF2-GDP accompanied by joining of the 60S ribosome subunit, allowing protein synthesis to commence [117].

The mTOR pathway promotes translation by stimulating assembly of the eIF4F complex [5]. One of the major mTORC1 downstream targets is inhibitory phosphorylation of eIF4E-binding proteins (4EBPs). This releases 4EBP from binding eIF4E and enables interaction between eIF4E and eIF4G, thereby enhancing eIF4F complex formation and 5′ cap-dependent translation. In the context of ageing, and well-investigated in *C. elegans*, the down-regulation of eIF2β/iftb-1, eIF4G/ifg-1, and eIF4E/ife-2, all extend lifespan in worms [118]. In a large RNAi screen for long-lived mutants, 2700 genes were targeted in worms, and one of the prominent categories consisted of translation initiation factors: eIF3/eif-3, eIF4A/inf-1, and eIF1B/eif-1 [119]. This and other experiments strongly suggest that reducing translation by down-regulation of a variety of initiation factors is an anti-ageing intervention [118,119,120,121,122,123]. In *Drosophila*, 4EBP is required for lifespan extension under dietary restriction, and it was proposed that when 4EBP is activated, this replicates condition of mTOR downregulation and low nutrients, and favours translation of mitochondrial genes with shorter and less structured 5′ UTRs [124]. When 4EBP is over-expressed in the muscles, it preserves muscle function and extends lifespan [125]. This illustrates that numerous translation initiation genes are good anti-ageing targets.

Another critical downstream target of mTORC1 that is implicated in ageing is S6K. S6K is phosphorylated and activated by both PDK and mTORC1 [126]. In all organisms tested, from SCH9 mutants in *S. cerevisiae* [127], S6K/rsks-1 RNAi in *C. elegans* [118], S6K^KQ^ dominant negative overexpressor flies [128], and to S6K1 null mice [129], deleting or down-regulating of S6K results in longer lifespan. S6K is one of the kinases phosphorylating ribosomal S6 protein, and its other translation targets include eIF4B and eEF2K, suggesting S6K can regulate both initiation and elongation step of protein synthesis [126]. Intriguingly, in mouse cells from S6K1^-/-^S6K2^-/-^ double knockout, global translation as measured by polysomal profiles was not affected [130]. However, majority of ribosomal biogenesis (RiBi) genes were transcriptionally down-regulated in S6K1 and S6K2 double knockout cells [131], convincingly showing a role of S6K in transcriptional regulation of nucleolar factors involved in ribosome maturation and assembly [131]. S6K has many other roles and targets that are involved in mRNA processing, protein folding, metabolism and cell survival some of which could be important in health and ageing [126,132]. For instance, under high mTOR signalling, S6K mediates negative feedback to insulin receptor substrate 1 (IRS1). When IRS1 is phosphorylated by S6K and mTOR, it is degraded and cells become unresponsive to insulin [133]. Interestingly, S6K1 null mice are resistant to diet induced obesity [134], and their transcriptomic changes resemble transcriptome under caloric restriction and AMPK activation by AICAR [129]. This suggests S6K as a promising drug target for health benefits.

Another layer of mTOR translation regulation and coordinated translational control stems from its effect on positively regulating translation of mRNA having 5′terminal oligopyrimidine (5′TOP) motif. Such motifs are found in components of translation machinery, more precisely ribosomal proteins and eEF2 [13,135]. Ribosomal profiling experiments showed that these 5′TOP mRNAs are major targets for translation down-regulation upon mTORC1 and mTORC2 inhibition using Torin 1 [136,137]. A polysome profiling study on the other hand, found many mRNAs without 5′TOP motif to be sensitive to mTOR pathway inhibition as well, such as cyclins and c-Myc [138]. This discrepancy is attributed to technical details between the two methods and better sensitivity of polysome profiling to detect small changes in translation. When ribosome profiling is used, greater sequencing depth may be required to detect less abundant mRNAs that are being translated [139,140]. Precise underlying mechanism of mTOR-mediated 5′TOP translation remains uncertain, and competition of eIF4F with a repressor protein such as Larp1 for 5′TOP motif is being proposed [140,141]. Because nearly all ribosomal mRNAs have 5′TOP motif, their mTOR-mediated regulation is interesting from a healthy ageing perspective as it is well established that ribosomal mutants live longer [142]. In worms, when rps6, rps11, and rps22, are down-regulated by RNAi, lifespan is extended [118]. In yeast, replicative lifespan of 107 ribosomal gene deletion strains showed that the large 60S subunit mutants are long-lived, a mechanism that resembles dietary restriction involving a nutrient-responsive transcription factor Gcn4, an ATF4 homologue [143]. Another argument for the importance of translation in ageing comes from the results of a deletion library screen, consisting of 4698 *S. cerevisiae* mutants. In this screen ribosome, mitochondrial translation, and tRNA export were the most significant functional categories among long-lived mutants [144]. It will be interesting to examine these different long-lived ribosomal mutants in light of the discovery that cells may possess heterogeneous ribosomes that can dictate translation of specific mRNAs [145].

In sum, there is an abundant evidence of the anti-ageing effect of down-regulation of translation. Long-lived mutants are found among different components of the translation machinery [146], and the mTOR pathway is the principal regulator of protein synthesis. Rapamycin, one of the first anti-ageing drugs discovered, inhibits completely mTORC1 pathway in budding yeast, while in mammals it inhibits less well some mTORC1-dependent processes, but its prolonged inhibition may also inhibit mTORC2 complex through interference with the complex assembly [2,5]. Based on numerous long-lived translation mutants, many other drugs targeting translation machinery are predicted to improve longevity and health [147].

A key question is: Why are different translation mutants long-lived [148]? Ageing is accompanied by reduction of translation, likely as a compensation for overall collapse of proteostasis, which consists of protein synthesis, protein folding and degradation [146,149]. Several hypotheses have been put forward, such as that reduced translation may be an anti-ageing intervention that saves cellular energy resources that could then be invested in cell maintenance. Reduced translation means protein folding and degradation machinery become more available to deal with damaged cellular components. Another interesting possibility is that under reduced translation, a different set of proteins are being synthesized and the proteome is more adapted for stressful conditions [148,150]. Finally, there is also a possibility that reduced translation could improve translation fidelity and lead to fewer protein errors [146,151], a hypothesis that was heavily debated in the past [152] but currently neglected. Some interesting correlative evidence suggest that animals having improved translation accuracy live longer, such as exceptionally long-lived naked mole rat [153]. In addition, hypoaccurate ribosomal yeast mutants have shorter chronological lifespan [154], and accuracy in yeast mitochondrial ribosomes positively correlates with longevity [155] (Figure 3).

Reduced translation is an important evolutionarily conserved anti-ageing intervention, and there is a plethora of different mutants and longevity mechanisms yet to be discovered in this field. To improve health in the elderly, different translation-modifying drugs could be tested for longevity and rapamycin is one of the first examples of how this approach may be successful.

## 4. mTOR and Autophagy

mTOR integrates information on growth factors, cellular nutrient status, energy level, and stress, in order to either promote growth when the environment is favourable, or trigger catabolic processes, such as autophagy, that degrade cellular components to obtain energy and building blocks [5,156] (Figure 4). Autophagy is, together with the proteasome, a major cellular degradation process, and the only process that can degrade entire organelles. There are three different types of autophagy, microautophagy, chaperone-mediated and macroautophagy, which differ in the way cargo is delivered to the lysosome; here we will refer to macro-autophagy as autophagy [157,158].

During autophagy, a portion of the cytoplasm is captured within the autophagosome, and then degraded by lysosomal hydrolases upon fusion with the lysosome, thereby replenishing the cytoplasm with amino acids, lipids, and nucleotides [159,160]. Two ubiquitin-like conjugation systems are critical for autophagosome formation. These systems enable covalent conjugation of Atg8, a ubiquitin-like protein, to phosphatidylethanolamine (PE) and its incorporation into the inside and outside membranes of the phagophore [161,162]. Atg8 is firstly cleaved by Atg4, and then lipidation is mediated by Atg7, the E1 enzyme, and Atg3, the E2 enzyme. Its localisation to the autophagic membranes is determined by the E3-like Atg5-Atg12 Atg16L complex, which is a second conjugation system driving the autophagy process. Autophagy proteins were discovered in yeast and more than 30 of them are now known, including the mammalian homologues of Atg8, LC3, and GABARAP proteins [163]. Atg8 interacts with different proteins that have LC3 interaction regions (LIR), enabling the autophagosome to be linked with specific cargo in a process of selective autophagy degradation, as opposed to general autophagy where cellular components are randomly recycled [164]. There is a growing number of LIR containing proteins, which are mainly autophagy receptors, but also components of the core autophagy machinery and of proteins being degraded. This is critical for precise elimination of certain cellular components, such as damaged mitochondria, in a process called mitophagy, bacteria by xenophagy, protein aggregates by aggrephagy, lipids by lipophagy and many other cellular macromolecules and organelles that can selectively be degraded [165]. For instance, ribosomal degradation mediated by starvation and mTORC1-inhibition via NUFIP1 autophagy receptor liberates both amino acids and ribonucleotides, which is important for subsequent mTOR reactivation [166]. Ribophagy has potential to be interesting in ageing research, given that many ribosomal mutants are long-lived across different model organisms [118,143] and that reduced ribosomal biogenesis is a well-established pro-longevity intervention [167].

mTORC1 inhibits autophagy by regulating different autophagy steps as well as lysosome biogenesis [168,169]. For instance, mTORC1 phosphorylates and inhibits Atg1, a homologue of human ULK1, as well as Atg13, both of which are part of the ULK1 complex together with Atg101 and FIP200 [170]. ULK1 has many phosphorylation sites, during starvation or mTORC1 inhibition, mTORC1 inhibitory phosphorylation is replaced by AMPK-mediated phosphorylation of ULK1 at different sites. AMPK is an energy sensor kinase that is active when the cellular AMP/ATP ratio increases and its phosphorylation of ULK1 promotes interaction and activity of the ULK1 complex [18]. mTORC1 also inhibits a PIK3C3/VPS34 complex, through inhibitory phosphorylation of Atg14, thereby providing an additional break on autophagy when growth conditions are favourable [171]. Moreover, mTORC1 is shown to block late stages of autophagy as well, by negatively affecting endosome to lysosome maturation through phosphorylation of UVRAG (UV radiation resistance associated), an important regulator of mammalian macroautophagy/autophagy [172].

Transcription of autophagy genes and lysosomal biogenesis factors requires the transcription factors TFEB and TFE3, and mTORC1-mediated inhibitory phosphorylation results in their cytoplasmic retention [43,173,174]. Interestingly, overexpression of mammalian TFEB orthologue HLH30 in worms leads to extension of lifespan that is dependent on autophagy [175]. This is one of the few direct genetic interventions that induces autophagy and promotes health and longevity [25,176]. Others include overexpression of the Atg8a protein in *Drosophila* neurons which diminished accumulation of damaged proteins in the brain [177] and a similar lifespan-extending effect of Atg8a is observed when it is overexpressed in fly muscles [178]. In *Drosophila*, it was shown that up-regulation of Atg1 can trigger the autophagy process [179] and this is also an anti-ageing intervention [180]. In mice on the other hand, it is Atg5 overexpression that extends lifespan and was accompanied by increased insulin sensitivity and improved motor function [181]. Despite numerous autophagy proteins, only some of them when overexpressed can trigger the entire process and extend lifespan. Additional evidence for the importance of autophagy in ageing was provided by experiments in which autophagy inhibition blocks lifespan extension of long-lived mutants, such as the *C. elegans* insulin receptor *daf-2* mutant [176]. Given that accumulation of damage, damaged organelles, and overall loss of proteostasis is a hallmark of ageing, it is not surprising that autophagy activation has beneficial effects and that many pro-longevity interventions depend on increased autophagy [176]. Activation of the lysosomal pathway was also recently associated with clearance of protein aggregates in quiescent neuronal stem cells, resulting in their younger, healthier state [182]. However, while there are numerous studies that show enhancement of autophagy is beneficial for ageing, we should nevertheless be cautious because if autophagy flux is clogged during ageing, then further autophagy enhancement could be detrimental and autophagy inhibition could be beneficial instead [183].

Overall, there are several points of interaction whereby mTORC1 signalling inhibits autophagy under favourable growth conditions. Autophagy is continuously active at basal levels; however, mTORC1 inhibition provides a degradation boost, cleansing the cell of damaged organelles via autophagy, which is a particularly beneficial anti-ageing intervention. There are several anti-ageing drugs that exert their effect by increased autophagy, such as valproic acid, lithium, trehalose, spermidine, urolithin A, and mTOR inhibitor rapamycin [25,176,184]. Most of these drugs have additional effects, therefore finding compounds that can specifically enhance autophagy, and even more precisely ones that can boost the selective types of autophagy, would be of great interest. These drugs, under continuous or an intermittent regime, would be promising candidates for improving health during ageing.

## 5. mTOR, Human Ageing, and Cellular Senescence

The links of mTOR with pathologies such as metabolic syndrome and neurological problems has been reviewed elsewhere [5]. More work is required to directly link mTOR function with human organismal lifespan. Accumulating evidence from model organisms and human cell lines provides great promise for mTOR pharmacological interventions in human ageing. With mTOR deregulation and associated mutations being linked to different types of cancers [2,5] the large number of clinical trials in various phases that include rapalogues (a class of allosteric mTOR inhibitors that are derivatives of rapamycin) and other mTOR-related pharmacological interventions and inhibitors (clinicaltrials.gov) is not surprising. The complexity and breadth of connections of mTOR signalling pathway with other nutrient- and growth factor-responsive pathways imposes great toxicity risks when using high doses of inhibitors [5,185] and optimal dosages and durations of treatment are unknown factors [186]. The most common side effects are immunosuppression, hyperglycaemia and dyslipidaemia as well as interstitial pneumonitis [187]. Importantly, a study on human elderly patients showed that low doses of mTOR inhibitors could improve immune function [188].

Nevertheless, significant amount of work conducted in human cellular models has shown that mTOR inhibition supresses the senescence associated secretory phenotype (SASP), which can disrupt tissues and contribute to age-related pathologies, including cancer [187]. For example, mTOR regulates SASP by promoting IL1A translation, which in turn promotes NFKB1 transcriptional activity [187,189]. Additional experiments utilising engineered human fibroblasts that can be arrested in G1 for determination of their chronological lifespan (CLS, define as the time that postmitotic cells can remain viable) have shown the importance of mTOR in human cellular ageing [190]. Rapamycin in this model was able to increase CLS while the effect was related to the acidification of the medium similarly to yeast lifespan studies [190]. In the same model, exposure to hypoxia suppressed the mTOR pathway, prevented senescent morphology and loss of regenerative potential, maintaining reversible quiescence [191]. Finally, dual mTORC1/C2 inhibitors such as Torin1 and Torin2 were shown to suppress hypertrophy, senescent morphology and extend CLS of human fibroblasts even better compared to rapamycin, suggesting that at doses lower than anti-cancer concentrations, pan-mTOR inhibitors could be developed as anti-ageing interventions [192]. Research in delaying pathological aspects of human ageing and, therefore, increasing health span has proven hard and elusive. Data from diverse experimental systems including human cellular models provide the optimism that mTOR inhibitors could provide the desired solution [193]. The complexity of the mTOR signalling hub, its diverse inputs and outputs [194], as well as interface with multiple other signalling pathways while complicate the matter, present the opportunity of testing pharmacological combinations or repurpose already approved drugs. The Dog Aging Project [195], bridging the gap between invertebrate, small mammals and human subjects as well as increasing numbers of mTOR-related trials, mark an exciting era in biogerontology pointing to the cross-roads towards regulating human lifespan and health span.

## Figures and Tables

**Figure 1 genes-11-01043-f001:**
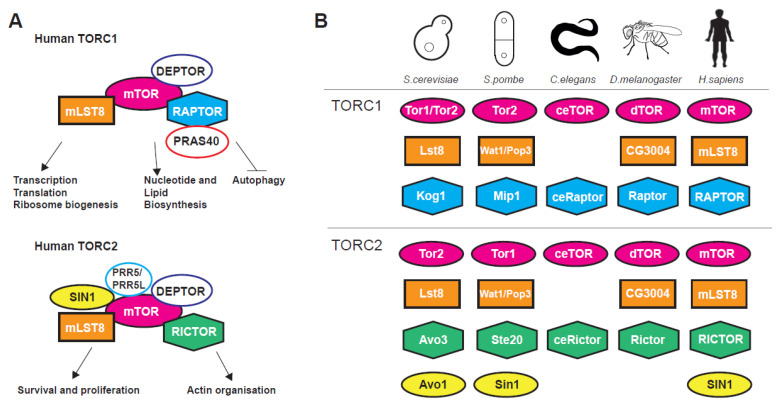
(**A**) TORC1 and TORC2 subunits and processes controlled by the complexes. (**B**) Target Of Rapamycin Complexes (TORC1 and TORC2) subunits in various model organisms used in ageing research.

**Figure 2 genes-11-01043-f002:**
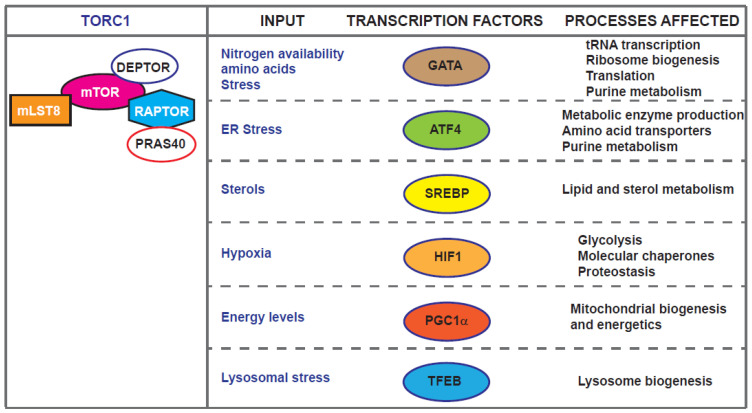
Transcription factors downstream of TORC1, input linked to their action and cellular processes affected.

**Figure 3 genes-11-01043-f003:**
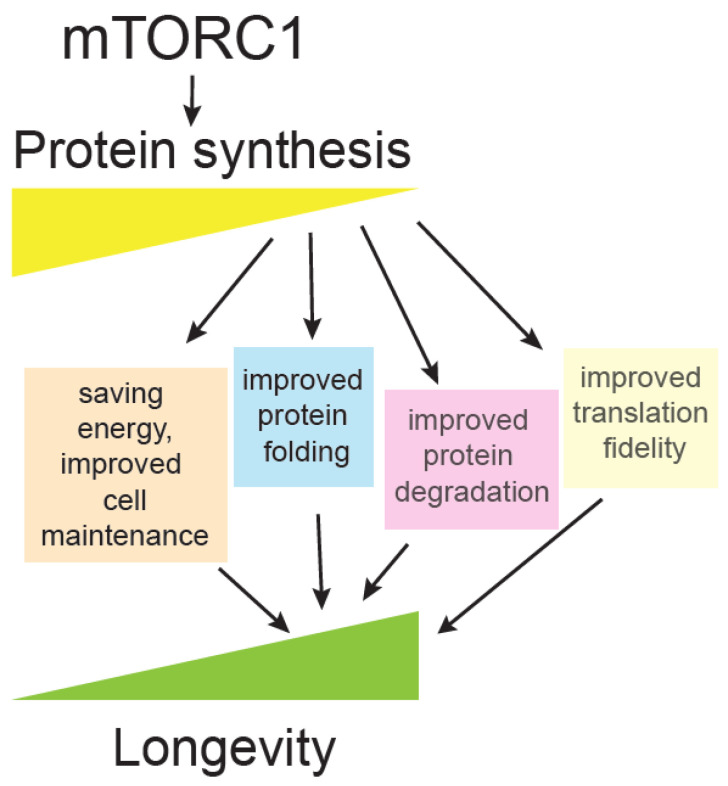
mTORC1 promotes protein synthesis. Decreased protein synthesis correlates with extension of lifespan and many mutants affected in translation initiation, elongation, and ribosomal proteins are long-lived. Some of the proposed mechanisms whereby less translation could promote longevity are illustrated.

**Figure 4 genes-11-01043-f004:**
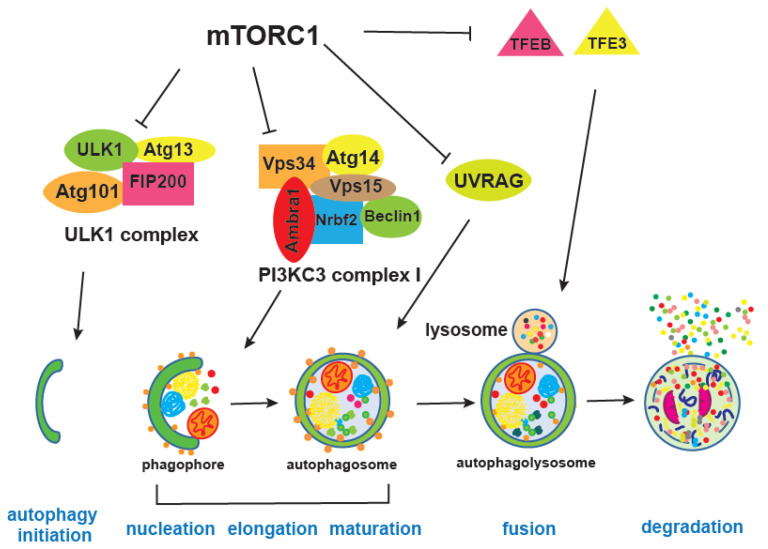
mTORC1 inhibits various steps of autophagy. mTORC1 phosphorylates and inhibits ULK1 and Atg13 which are part of the ULK complex, thereby blocking autophagy initiation step. mTORC1-mediated phosphorylation of Atg14, AMBRA1, and NRBF2 from the PI3KC3 complex I, which is one of the PI3KC3 complexes, negatively regulates autophagosomal nucleation step. mTORC1 also affects maturation of autophagosome through inhibitory phosphorylation of UVRAG. TFEB and TFE3 are transcription factors critical for lysosomal biogenesis and expression of autophagy genes. Different steps in the autophagy process are illustrated and denoted in blue. Bold letters in different complexes denote proteins that are phosphorylated by mTORC1.

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
