# Peer review of "The Target of Rapamycin Signalling Pathway in Ageing and Lifespan Regulation"

_genes, 2020, doi:10.3390/genes11091043_

Round 1

Reviewer 1 Report

Since the discovery of the longevity effect of rapamycin in various model organisms, the regulation of lifespan by the mTOR signaling pathway has been extensively studied. In the current manuscript, the authors review the current knowledge of the mechanisms by which mTOR signaling controls longevity and aging, particularly focusing on fundamental cellular processes, such as transcription, translation, metabolism and autophagy. In addition, the article includes suggestive comments on potential therapeutic approaches to influence longevity and aging-related diseases. The manuscript is readable, and the figures serve as excellent visual aids.

1) Figure 1B indicates S. pombe TORC1 contains Tor1/Tor2, probably based on a paper by Hartmuth and Petersen (JCS 122, 1737). However, this has not been confirmed by other groups, and Ikai et al. (Open Biol 1, 110007) argued against it. Thus, it would be less controversial if Tor1 is deleted from the S. pombe TORC1 components in the figure.

2) Inattentive errors abound throughout the manuscript, including those listed below. Careful proofreading would be highly appreciated before submission.

  • Line space is required between lines 64 and 65.
  • Line 70: “REPTOR-Binding Protein (P)” should be “REPTOR-Binding Protein (REPTOR-BP or BP)?”.
  • Line 88: the period after “[50]” should be a comma?
  • Line 196, The abbreviation “DR” should be defined.
  • Indent line 301.
  • Only the subheading in line 328 is capitalized.

Author Response

We thank the reviewer for the useful input and advice. Please find below a point-by-point answers (in bold) to comments (in normal writing):

1) Figure 1B indicates S. pombe TORC1 contains Tor1/Tor2, probably based on a paper by Hartmuth and Petersen (JCS 122, 1737). However, this has not been confirmed by other groups, and Ikai et al. (Open Biol 1, 110007) argued against it. Thus, it would be less controversial if Tor1 is deleted from the S. pombeTORC1 components in the figure.

We thank the reviewer for this comment. We have amended the figure with one in which Tor1 is now deleted from S. pombe TORC1.

2) Inattentive errors abound throughout the manuscript, including those listed below. Careful proofreading would be highly appreciated before submission.

We apologise for such errors. We have corrected errors including the ones pointed by the reviewer:

  • Line space is required between lines 64 and 65

This is now inserted. 

  • Line 70: “REPTOR-Binding Protein (P)” should be “REPTOR-Binding Protein (REPTOR-BP or BP)?”.

We have replaced this with "REPTOR-BP".

  • Line 88: the period after “[50]” should be a comma?

We have removed the full-stop after [50] in Line 88. This reads better now and we thank the reviewer.

  • Line 196, The abbreviation “DR” should be defined.

We have now defined the term.

  • Indent line 301.

We have inserted the indent as required for a new paragraph.

  • Only the subheading in line 328 is capitalized.

The subheading in line 328 is now in normal fonts to be in accordance with the rest of the manuscript. 

Reviewer 2 Report

The review entitled “The Target of Rapamycin signalling pathway in ageing and lifespan regulation” is a well-composed informative review for a wide range of readers. It is well written and nicely presented. In my opinion, this MS, when published, will be a worth review to read for a wide range of readers.

My only request to the authors is to embellish the MS with a section specific for Homo sapiens, which seems to be lacking. Considering the complex relationship between the growth factor signaling network and senescence in human beings, a section specific for Homo sapiens, in the context of mTOR, will be a reader’s delight. An additional section will not only complete the entirety of the review from a comparative aspect but will also justify authors’ claim that “A good understanding of the pathway’s outputs and connectivity is paramount towards our ability for genetic and pharmacological interventions for healthy ageing and amelioration of age-related disease.” If I am reading it correctly, authors meant “…..genetic and pharmacological interventions for healthy ageing and amelioration of age-related disease” in human beings.

Here are a few suggested essential articles which will be useful to include for the purpose:

  • mTOR as Regulator of Lifespan, Aging, and Cellular Senescence: A Mini-Review;            PMID: 29190625
  • mTOR at the nexus of nutrition, growth, ageing and disease:                                                          PMID: 31937935
  • mTOR is a key modulator of ageing and age-related disease;                                                                     PMID: 23325216
  • Growth Factor and Signaling Networks;                                                                                    doi:10.1016/B978-0-12-374984-0.00664-1
  • mTOR and Aging: An Old Fashioned Dress.;                                                                                                              PMID: 31174250
  • Regulation of the mTOR complex 1 pathway by nutrients, growth factors, and stress; PMID: 20965424
  • Rapalogs and mTOR inhibitors as anti-aging therapeutics.                          doi:10.1172/JCI64099

Minor typos:

Line # 328: “m” for mTOR in the place of capital “M.”

Author Response

We thank the reviewer for the useful input and comments. Below is a point-by-point answers (in bold) to the reviewer's comments (in normal writing). 

My only request to the authors is to embellish the MS with a section specific for Homo sapiens, which seems to be lacking. Considering the complex relationship between the growth factor signaling network and senescence in human beings, a section specific for Homo sapiens, in the context of mTOR, will be a reader’s delight. 

We have inserted a paragraph at the end of the review that includes aspects of mTOR in human ageing, senescence and age-related diseases. 

Line # 328: “m” for mTOR in the place of capital “M.”

We have corrected this together with the small capital format of the subheading to harmonise with the rest of the manuscript.